# Managing Apparent Loss and Real Loss from the Nexus Perspective Using System Dynamics

**Seo Hyung Choi [1], Bongwoo Shin [1] and Eunher Shin [1,2,*]**

[1] UNESCO International Centre for Water Security and Sustainable Management (i-WSSM),
Daejeon 34045, Korea; seohyung.choi@unesco-iwssm.org (S.H.C.); sbw7109@unesco-iwssm.org (B.S.)

[2] K-Water Institute, Daejeon 34045, Korea

[*] Correspondence: shin2@kwater.or.kr; Tel.: +82-42-349-0007

**Abstract:** When water utilities establish water loss control programs, they traditionally focus on apparent loss rather than real loss when considering economic feasibility in the water sector. There is an urgent need for new management approaches that can address complex relationships and ensure the sustainability of natural resources among different sectors. This study suggests a novel approach for water utilities to manage water losses from the water-energy (WE) Nexus perspective. The Nexus model uses system dynamics to simulate twelve scenarios with the differing status of water loss and energy intensities. This analysis identifies real loss as one of the main causes of resource waste and an essential factor from the Nexus perspective. It also demonstrates that the energy intensity of each process in the urban water system has a significant impact on resource use and transfer. The consumption and movement of resources can be quantified in each process involved in the urban water system to distinguish central and vulnerable processes. This study suggests that the Nexus approach can strongly contribute to quantifying the use and movement of resources between water and energy sectors and the strategic formulation of sustainable and systematic water loss management strategies from the Nexus perspective.

**Keywords:** WE Nexus; urban water system; water loss management; real loss; apparent loss; energy intensity; system dynamics

## 1. Introduction

Mounting water-related global challenges, such as climate change, water scarcity, escalating water demand due to population growth and urbanization, water deterioration, and aging infrastructure, are placing more pressure on water utilities than ever before. Water losses in water networks remain a significant concern worldwide, as it causes water wastage, technical burdens, water contamination, and revenue loss [1]. Since 2000, non-revenue water (NRW) management has been given high priority by policymakers, government officials, utility managers, and professional groups working within the water sector for efficient resource utilization, the commercial viability of water utilities, and the improvement of service provision [2].

Water utilities establish water loss control strategies and design programs by weighing economic, technical, social, and environmental aspects [3–12]. The main underlying principles for the strategy include four aspects. First, the strategy should be holistic because reducing NRW cannot be solved through a single project [4]. Multiple activities, such as water audits, the establishment and management of district metered areas (DMAs), leakage detection and repair, and pressure management can be categorized into modules depending on local circumstances. Second, water loss control programs must be flexible and customized to water supply systems' specific needs and characteristics [13]. Appropriately tailored counter-activities should be selected based on the types and volumes of leakage and the costs of the techniques implemented to reduce specific leakage

components. Therefore, it is essential that water utility managers conduct appraisals of a network's physical characteristics and assessments of current operational practices to understand the reasons why, how, and where water is being lost [14]. Third, water loss programs must be viewed from a long-term perspective that must continuously reach the economic level of leakage and maintain low levels once initial progress is made [15]. It must be emphasized that although early gains can be made in reducing NRW, there is no shortcut to strategizing the long-term sustainability of reduced water loss. Management procedures related to a utility's organization, procedures, and human resources must be revised to achieve permanent results. Lastly, all water utilities should preferentially set their sights on apparent loss (AL) caused by meter inaccuracies, data handling and billing errors, and unauthorized consumption (e.g., meter tampering and water theft) rather than real loss (RL), such as background leakage and pipe burst leakage [4]. The recovery of AL is possible with little effort at a relatively low cost, and will directly improve the water utility's financial position, especially at the beginning of an NRW reduction program [16]. However, it requires sustained management commitment, political will, and community support. These principles are universally accepted and applied by water utilities worldwide; however, one of the current issues is that strategies to address natural resources management have historically been characterized by sectoral approaches and isolated policy responses [17]. If a substantial amount of the water produced is lost through leakages and never reaches end consumers, it also means that the energy used to treat and distribute the water is wasted. Greenhouse gases (GHGs) are also emitted through energy generation as well as water and wastewater treatment processes. Therefore, new management approaches are needed to address complex relationships and ensure the sustainability of natural resources by interpreting the interactions and feedback among different sectors related to water.

"Nexus thinking" was first conceived by the 2011 World Economic Forum to promote the conception of inseparable links between the use of resources to provide fundamental and universal rights to food, water, and energy security [18]. Although various researchers and organizations have suggested the Nexus definition with contrasting interpretations in differing sectors and contexts [19–22], no consensus on the definition of the Nexus has been reached. What is certain is that the ultimate goal of the Nexus approach is to identify potential synergies and minimize trade-offs between the sectors. In the water sector, the Nexus approach has emerged in the form of Integrated Water Resources Management (IWRM), which emphasizes a multifaceted approach of addressing the resource [23]. Currently, IWRM has been implemented in several countries to balance water allocation for energy (i.e., hydropower generation), food demand (i.e., irrigation), and environmental protection (i.e., river flow maintenance). However, the IWRM considers water as the primary component, while other sectors are dependent [19]. In overcoming the limitations of IWRM, the magnitude of considering water-related problems from the Nexus perspective is gradually increasing. Specific to Agenda 2030, three of the seventeen sustainable development goals (SDGs), such as zero hunger (SDG 2), clean water and sanitation (SDG 6), and affordable and clean energy (SDG 7), are directly related to the water, food, and energy sectors [19]. The Nexus has been identified as a helpful approach for quantifying and assessing the interactions between the different goals [24,25]. The effects that the fulfillment of one goal may have on the realization of others can also be estimated. Various approaches, frameworks, and methodologies that are largely borrowed or adapted from conventional disciplinary approaches have been proposed for analyzing the Nexus. Questionnaire surveys [26–29], input–output analyses [30–36], cost–benefit analyses [5], lifecycle assessment [37–40], system dynamics (SD) [41], agent-based modeling [42], statistical applications [43–45], and mechanistic modeling [46–50] are widely used. In addition, innovative tools, such as CLEW3 [51], MuSIASEM [52], GAEZ-WEAP-LEAP [53], and MESSAGE [54], have all been developed and proposed. Unfortunately, each Nexus case is unique, and no general and comprehensive Nexus modeling approach fits modeling and

quantifying the interlinkages between sectors for all situations [27,55]. Different methodologies have differing data requirements, benefits, and limitations and only operate at particular geographical scales [56]. Therefore, it is crucial to select an appropriate modeling method according to the understanding of distinct temporal and spatial scales, interlinkages, actions of various stakeholders from each sector, and data availability.

The processes of providing drinking water, removing sewage, and draining storm water constitute an urban water cycle (UWC) or an urban water system (UWS) [57,58]. Water utilities are primarily responsible for UWS processes. Urban water system management is rapidly becoming more about the management of complex systems rather than just a few isolated issues, and the dynamic complexity presents challenges for UWS management. Operational and maintenance processes in a UWS have been identified as the most energy-intensive activities because energy consumption is directly related to both the quantity and desired qualities required by consumers [59,60]. Energy consumption has also been proved to have a positive correlation with carbon emissions [27]. Therefore, several Nexus approaches have been introduced into UWS to optimize resource management and reduce the greenhouse effect in recent years [61,62].

System dynamics is a well-established methodology based on the system concept and system theory that quantifies system behaviors with complex feedback for more accurate projections [63,64]. The method was proposed in the early 1960s, and has been widely adopted to analyze a diverse range of problems. Multiple researchers have demonstrated the suitability of SD in UWS management, as it is suited to modeling the interconnected and interdependent cause-and-effect chains in water systems [65–67]. The SD approach allows for a practical trade-off analysis of multi-scenario and multi-attribute to be conducted to facilitate the relative comparison of several alternative management strategies over time [68]. Urban water system planners, municipalities, and managers can then thoroughly assess options and meet the challenges of policy formulation and decision-making for sustainable development.

This study proposes a novel approach for water utilities to manage water losses from the Nexus perspective. Water utilities can implement this approach and overcome the limitations of traditional water loss management, which focuses only on the water sector. Water utilities can quantify each water loss management activity's synergies and trade-offs in the water and energy sector and establish suitable water loss management strategies and programs from a Nexus point of view. To this end, a water and energy (WE) Nexus model was built, and a scenario analysis was performed. The WE Nexus model that includes intake, conveyance, water treatment, transmission and distribution, sewage collection, wastewater treatment, and discharge processes was constructed using SD. Twelve simulated scenarios were examined, which were based on the current status of urban water losses and urban energy intensity. The intention of this process is to support water utilities' decision-making according to specific circumstances of water loss and intensity. In addition, water consumption, energy use, and $CO_2$ emissions are quantified in each process of UWS to identify the most energy-intensive processes for mitigation.

The remainder of this paper is organized into four sections: Section 2 presents the materials and methodologies used in the study; Section 3 details the results; Section 4 offers a discussion of the findings; and, finally, Section 5 presents conclusions and potential future research directions.

## 2. Materials and Methods

### 2.1. Modeling Scope

Developing a practical model and providing meaningful insights requires selecting the appropriate method, clarifying the scope, developing and integrating the model's diverse components, and examining potential interventions (e.g., new technological and resource development alternatives) that have not previously been adopted in the sector, which are central to Nexus analyses [69,70]. In this study, a model was constructed using

SD that was deemed suitable for expressing the causal relationships between established UWS variables. The model scope, structure, and the energy intensity values that are important parameters in the model are described in detail in Sections 2.1–2.3, respectively. The intervention to derive the direction of water loss management from the Nexus perspective and the corresponding scenarios are discussed in Section 2.4.

The Nexus approach takes various forms depending on the scope of the system under examination. It is critical to define and clarify system scope during the model building phase to advance a practical and goal-oriented analysis. The sectors, geographic scale, application of the model results, interaction assistance, and indicators established in this study are presented in Table 1.

**Table 1.** Modeling the scope of a water-driven water and energy Nexus for an urban water system.

| Items | Details |
| --- | --- |
| Sector | Water, energy |
| Geographic scale | Urban level |
| Application of model results | Understanding the Nexus |
| Interaction assumption | One-way impact analysis (water → energy) |
| | Building a water-driven Nexus |
| Indicators | Water footprints, total energy use, and total $CO_2$ equivalent emission |

Although the implementation of the Nexus approach typically requires linking different sectors, it is unnecessary to build a Nexus model that involves all policy domains, such as water, energy, food, land, environment, climate, and ecosystems. The interconnection between any two of these sectors can constitute a Nexus examination. Therefore, it is vital to preferentially select sectors that are most related to UWS. The water-energy Nexus has been introduced into UWS to improve the scarce resources and greenhouse effect in recent years [62]. Understanding the Nexus of energy and water provides opportunities to enhance water and energy supply sustainability and minimize energy and water consumption through regulatory cooperation at higher institutional levels. The water-energy-carbon Nexus has been researched to assist municipalities, urban developers, and policymakers in making informed decisions for reducing water consumption, energy use, and carbon emissions in UWS [71–73]. The total input, distributions, consumptions, and output of water, energy, and carbon flow are examined stage by stage in the water system [61]. In this study, water, energy, and carbon emissions were considered essential elements of urban water sustainability, as these elements are interconnected and have complex interactions. Energy is used in every stage of the water system, including water intake, water treatment, distribution, end-use, and wastewater treatment [74–79]. Investigating the relationships between energy and water consumption can reveal insights for reducing the GHG emissions associated with urban water sectors [27]. In this study, water and energy sectors were selected for water, energy, and carbon flow analysis, referring to this interconnection as the WE Nexus.

Nexus assessment can be conducted at the various geographical scales of household, city, regional, national, transboundary, and global levels, and it is essential to identify the scale at which a Nexus problem should be addressed. The constructed scale has a significant impact on identifying stakeholders and determining the data required during model building [69]. Nexus analyses are often conducted at regional or national levels due to data availability and national-level policy goals. This study sought to analyze water, energy, and carbon flows; thus, it is appropriate to set the geographic scale of the Nexus at the city level.

Stages of Nexus model results can be classified into the three categories of understanding, governing, and implementing [80,81]. In the understanding stage, researchers solely focus on the quantitative analysis between sectors to uncover linkages and identify

critical challenges, risks, and/or opportunities. The model is constructed to guide institutional and policy responses to the problems revealed at the governing stage. In the most complicated stage—the implementation stage—the study is designed to guide policy and/or technical interventions to improve the efficiency and effectiveness of resource use. This study monitored interlinkages in the WE Nexus to accurately and effectively identify and understand the potential synergies and trade-offs between the three sectors.

Nexus system analysis can reveal the internal features of a coupled system by capturing the interactions between different sectors. The interactions between different sectors can be classified into one-way impact analyses or interactive impact analyses [82]. One-way impact analyses are based on unilateral relationships and are a targeted and straightforward approach to uncover how changes in one specific sector affect other sectors as well as to facilitate a preliminary assessment of associated trade-offs. This approach has recently been widely employed to analyze the feasibility and impact of new technology applications within the water sector on others, as disruptive and exponential technologies provide new means for addressing current water-related challenges [81]. Interactive impact analyses consider mutual relationships and feedback loops, are complex and holistic, and are well-suited for achieving a more comprehensive assessment and identifying the originating factors. This study adopted a one-way impact analysis and developed a water-driven Nexus model to analyze the effect of 12 water loss control strategies on the energy sector.

Measurable variables were used as indicators to represent, quantify, and capture the systems' overall characteristics, regardless of their complexities. Indicator-based methods enabled a comparative assessment and benchmarking by rendering system performance variables in a uniform and standardized format. The indicator-based assessment and benchmarking of the Nexus also allowed for the incorporation of several aspects into the analysis and minimized biased assessments at study sites. Water footprint [$m^3$], total energy use [kWh], and total carbon emission [$kgCO_2eq$] were used to describe the resource use and transfer of the water and energy sectors, respectively.

### 2.2. Model Structure

The conceptual framework for the working mechanism of UWS was established by selecting the crucial components of the model and identifying the essential causal relationships and feedback among the components to develop the UWM Nexus model using SD. As shown in Figure 1, the key stages of the UWS are usually divided into drinking water processes, customer use, and wastewater processes. In the UWS, raw water from sources, such as lakes, rivers, and underground aquifers, is extracted and conveyed to the water treatment plant. The delivered raw water is then treated to be potable and palatable according to the drinking water standards of the water treatment plants, and then the water utilities distribute drinking water to customers. In the customer-use stage, the water supplied is used for residential, commercial, and industrial purposes. Sewage is collected from customers and transferred to wastewater treatment plants. The contaminants in wastewater are removed by wastewater treatment plants for reuse or discharge into the natural water cycle. In some countries, water reuse is considered an alternative for the irrigation of public spaces, agriculture, potable use, and groundwater replenishment.

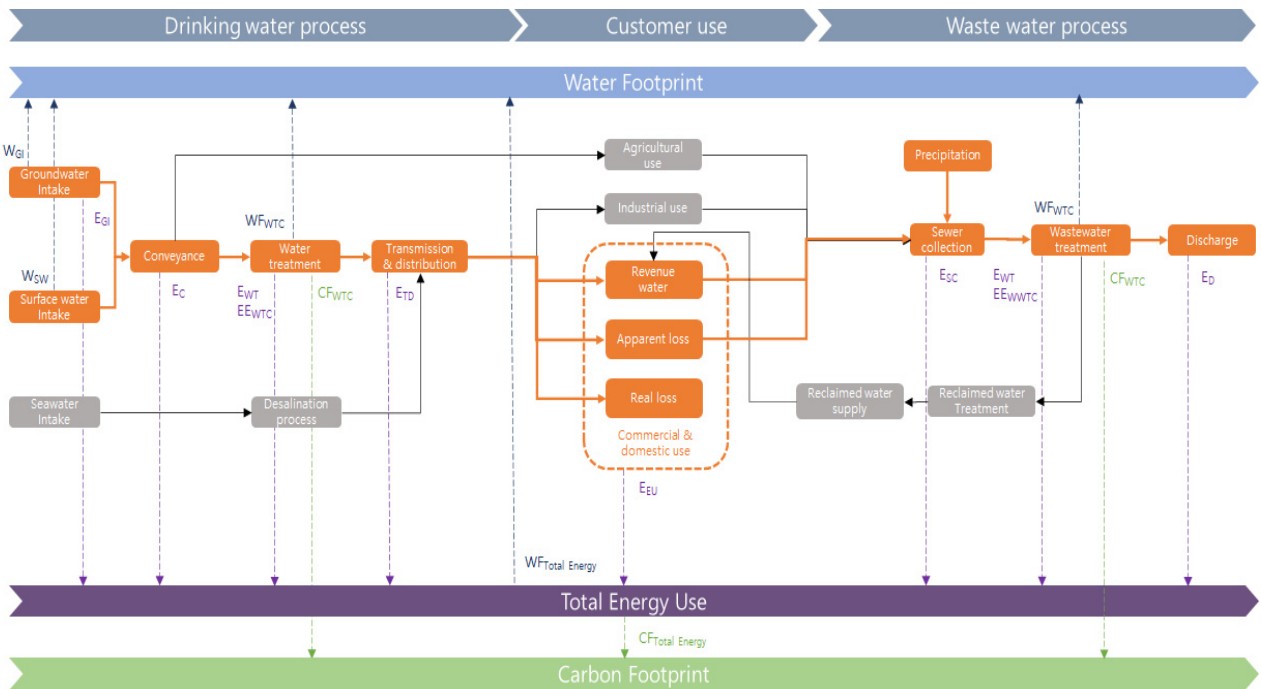

**Figure 1.** Components and interlinkages of the water and energy Nexus model.

Previous UWM SD models [58,65,67] have included intake, desalination, conveyance, water treatment, transmission and distribution, water use (e.g., agricultural, industrial, commercial, and domestic use), sewer collection, wastewater treatment, water reuse, and discharge as vital processes. All of these processes are included in the WE Nexus model constructed for this research, with the exception of desalination, agricultural and industrial use, and water reuse. As shown in Figure 1, the processes considered are marked in orange, and those that are not included are marked in gray. Due to the lack of water resources, some developed countries (e.g., Israel and Singapore) consider desalination and water reuse as alternative water resources; however, integrating these non-conventional water resources is not practiced worldwide because of prohibitive costs and considerable energy consumption. Agricultural and industrial consumption does not account for a substantial portion of water use in cities. Additionally, the treatment processes and supply schemes for agricultural and industrial water differ from those of commercial and domestic water; raw water is primarily used for agricultural water, and treated water without disinfection is used for industrial water. In this study, to analyze how the NRW affects the WE Nexus, commercial and domestic use were integrated and classified into revenue water, AL, and RL.

Previous research has determined the essential causality and feedback for each process for research purposes [36,58,65–67]. Among these studies, the interlinkages of the three representative models are summarized in Table 2. Sometimes, simplified SD models are needed to increase the quality and understanding of the models. The number of components must be appropriately large enough to capture the necessary details, yet small enough to be manageable. Weak feedback and stock variables embedded in an SD model's loop can be excluded. Simple cause–effect relationships and feedback were selected to analyze the effect of water loss in UWS for this study. Accordingly, the model variables, relational expressions, and units are described in detail in Table A1 in Appendix A.

**Table 2.** The interlinkages in the urban water system within the literature.

| | | [58] | | [67] | | [65] |
|---|---|---|---|---|---|---|
| Water resource | - | Surface water (−) | -<br>- | Surface water (−)<br>Ground water (−) | - | Surface water (−) |
| Intake/Conveyance & transmission | | O | | X | | X |
| Water treatment | - | Chemicals for water treatment (+) | | X | | X |
| Distribution | | O | | X | | X |
| Groundwater recharging | | X | -<br>-<br>-<br>- | Returned water (−)<br>Natural recharge (−)<br>Natural discharge (+)<br>Water extraction (+) | | X |
| Other water resources | - | Reclaimed water | | O | | O |
| Population | -<br>- | Population growth rate (−)<br>Residential water use (+) | -<br>-<br>-<br>-<br>- | Birth (birth rate) (−)<br>Death (death rate) (−)<br>Migration (migration rate) (−)<br>Landscape demand (+)<br>Domestic demand (+) | -<br>-<br>-<br>- | Birth rate (−)<br>Death rate (+)<br>Immigration rate (−)<br>Emigration rate (+) |
| Water losses | - | Water loss | -<br>- | Leakage rate (−)<br>Loss reduction (+) | - | Distribution leakage rate (−) |
| Water use | -<br>-<br>-<br>-<br>-<br>- | Residential water use (−)<br>Commercial water use (−)<br>Institutional water use (−)<br>Industrial water use (−)<br>Golf and parks water (−)<br>Agricultural land (−) | -<br>-<br>-<br>- | Industrial demand (−)<br>Landscape demand (−)<br>Domestic demand (−)<br>Other demand | -<br>- | Water-only end uses (−)<br>Water-energy end uses (−) |
| Sewage collection | -<br>-<br>-<br>- | Infiltration inflow (−)<br>Reclaimed water (+)<br>Chemicals for WW treatment (+)<br>Biosolids transportation (+) | | X | | X |
| Wastewater treatment | | O | | X | | X |

Monitoring the multiple interlinkages between water and energy sectors is crucial to understanding potential synergies and trade-offs. Quantification is also needed to elicit a more comprehensive understanding of the numerous interlinkages to facilitate improved strategies and decision-making [83]. To quantify resource interlinkages in the model, water footprint, total energy usage, and carbon footprint were considered as evaluation indicators, respectively, as shown in Figure 1. The water footprint was calculated as the sum of the amount of groundwater ($W_G$) or surface water abstracted ($W_{SW}$) during UWC, the amount consumed during the production of chemicals used in water treatment ($WF_{WTC}$) and wastewater treatment processes ($WF_{WWTC}$), and the amount of water required to produce the energy consumed during UWS ($WF_{TotalEnergy}$). Total energy consumption is the total amount of energy used in groundwater intake ($E_{GI}$), water conveyance ($E_C$), water treatment ($E_{WT}$), water transmission and distribution ($E_{TD}$), sewage collection ($E_{SC}$), wastewater treatment ($E_{WT}$), and chemical production for water treatment ($EE_{WTC}$) and wastewater treatment processes ($EE_{WWTC}$). The carbon footprint was calculated by adding the amount of $CO_2$ equivalent ($CF_{Total\ Energy}$) emitted in the generation of energy used in UWS and in chemical manufacturing for water treatment ($CF_{WTC}$) and wastewater treatment ($CF_{WWTC}$).

### 2.3. Energy and Gas Intensity

The interactions in the WE Nexus are reasonably categorized in terms of resource use efficiency, including water intensity, energy intensity, and carbon equivalent. Water intensity [m³/kWh] is the volume of water consumed to produce a unit of energy. The rate of electricity consumption during water production is termed energy intensity and is expressed in units of kWh/m³. Most activities involving energy use and combustion produce $CO_2$ or other GHG emissions. The carbon equivalent [kgCO₂eq/kWh] is used to quantify carbon emission as a single unit. Energy intensity and carbon equivalent have a relatively more significant influence on model accuracy and reliability than water intensity, as demonstrated by the water-driven Nexus model explained in Section 2.1. The majority of carbon generation is also not directly related to water use, but to the production of energy. Therefore, energy intensity was used according to specific processes in UWS and water intensity and carbon equivalent were established as constant values of 0.02 m³/kWh and 0.25 kgCO₂eq/kWh in the model, respectively.

The level of energy required for each UWS process depends on the type and quality of the water source(s), topography, applied technology, and the efficiency of the water treatment and delivery system [84]. Many researchers have investigated energy intensity according to the technology applied for each process in UWS, and the summarized results are shown in Table 3.

**Table 3.** Energy intensity of each process in the urban water system.

| Process in UWS | Specific Technique | Energy Intensity (kWh/m³) | Reference |
|---|---|---|---|
| Intake | - | 0.0027~0.05 | [85,86] |
| Conveyance & transmission | - | 0.21~4.07 | [87–93] |
| Water treatment | Surface water & ground water | 0.01~16.4 | [33,79,86,89,94–113] |
| | Desalination | 0.36~68.69 | [114–117] |
| Distribution | - | 0.2~4.9 | [118] |
| Use | | 1.5~50 | [118] |
| Sewage collection | | 0 | - |
| Wastewater treatment | | 0.05~7.50 | [33,79,86,94,96–100,103,104,106–108,110,111,113,119] |
| Reuse | Centralized system | 0.72~3.8 | [87,89,120–124] |
| | Decentralized system | 1.7~4.5 | [119] |
| Discharge | | 0.02 | [119] |

In the intake process, the abstraction of groundwater is more energy intensive than the use of surface water. An investigation of intake energy for groundwater and surface water found that the energy use for groundwater abstraction is typically 27% higher than that of surface water [125]. The energy intensity for groundwater is mainly related to groundwater elevation and pump efficiency. The energy required for water conveyance, transmission, and distribution depends on pipe length, friction, network pressure, leakage rate, and topography. Considerable energy is consumed when water utilities pump raw water to water treatment plants or treated water to reservoirs and to long-distance consumers. Energy intensity and topography also have a strong relationship. The energy intensities of the water treatment step are directly and strongly affected by water quality and applied technologies. Groundwater treatment requires less energy than surface water because surface water includes more total dissolved salts [126]. Desalination demands more energy than conventional water treatment processes (i.e., groundwater and surface water); however, this gap is gradually narrowing with advances in related technologies. Energy consumption for end-use is comparatively higher than other processes in UWS due to water heating, swimming pools, washing machines, dishwashers, and cooking activity at domestic, commercial, and industrial levels. Among these factors, heating is the

most prominent influencing factor. It is reported that water heating accounted for 97% of energy consumption for water end-use in households in Australia [127] and 75% in the United States [22]. There are two types of sewer systems, being combined and separate sewer systems; a combined sewer system transfers surface run-off and wastewater together, while a separate sewer system carries them independently. In most cases, sewage is delivered through gravity, so energy is not consumed. The amount of energy consumed in wastewater treatment depends on water contamination and impurity types, treatment process types, discharge standards, and system operation efficiency. In particular, the amount of energy used in primary (e.g., screening, chemical treatment, grit removal, and sedimentation), secondary (e.g., aeration, stabilization, suspended growth, clarification, and membrane bioreactor), and tertiary (e.g., nitrification and de-nitrification) wastewater treatment stages showed significant differences. Tertiary treatment is the most energy-intensive process, while primary treatment is less energy intensive compared with the other two processes. The now clean, clear, and odorless reclaimed water can be allocated for the irrigation of public spaces, agriculture, cooling, and potable reuse. These processes require advanced technologies to meet acceptable water quality standards, so most are energy intensive.

### 2.4. Model Building, Calibration, and Simulation

The WE Nexus model for UWS using SD was constructed according to the scope, structure, and parameter values described in Sections 2.1–2.3. The commercial software, Vensim, was used to build the SD model. This study considered a virtual city as the subject of analyses. Its initial population, the population growth rate, and water demand per capita per day were established as 300,000 people, 0.001 L/month, and 100 L/person/day, respectively. The causal loop diagram of the model is presented in Figure 2. This model is water-oriented, and estimates the consideration of various alternatives of policy intervention and disruptive technology implementation concerning water use, energy use, and $CO_2$ emission.

An SD model can be calibrated, revised, and validated to support dimensional consistency, structural suitability, and historical consistency (i.e., through behavior pattern tests). Dimensional consistency refers to the assessment of whether the units of all variables set in the model are consistent with those calculated within the model. This evaluation can expose errors of the applied mechanisms, equations, and parameters, allowing modelers to accurately modify them. If the model behaves according to the rules of the world that it is attempting to simulate, the model is considered to be structurally suitable. Although it is not feasible to thoroughly assess all aspects of structural suitability, modelers should ensure that reality and the model results do not excessively deviate. A technique to compare the model's results with historical data (e.g., water consumption per capita, energy consumption per capita, and groundwater level) is the most widely used method in model calibration and verification. The model is preferentially calibrated using historical data other than the period used for validation. The calibrated model is then validated by using historical data. In this study, a historical consistency assessment was not used because the model was applied to virtually constructed cities. Model calibration and verification were performed with dimensional consistency and structural suitability. The commercial software used in this research includes a built-in unit check that confirmed the model's dimensional consistency. Structural suitability is also confirmed because the model was constructed referring to the general principles of UWS and understanding shaped by previous literature.

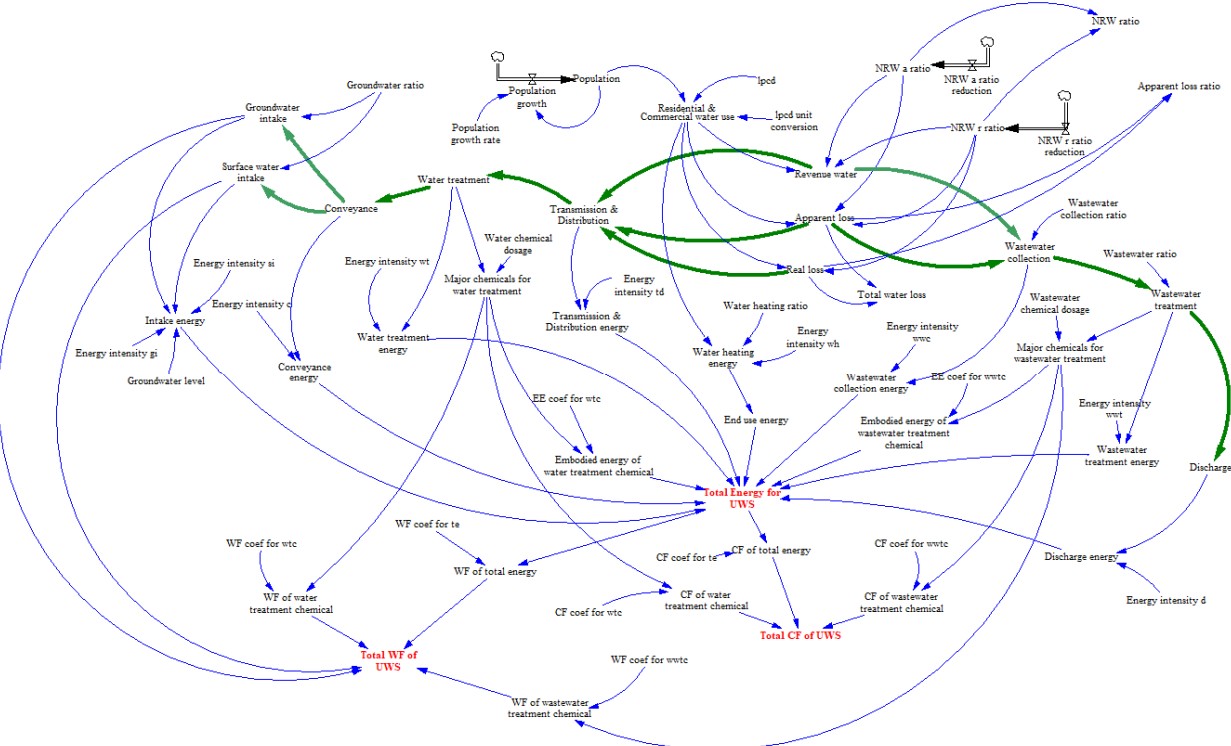

**Figure 2.** Causal loop diagram of the WE Nexus model for UWS.

All of the processes in UWS are interlinked and interdependent, functioning as a dynamic whole. Therefore, policymakers, urban developers, and water and wastewater utilities have multiple alternatives for maximizing synergies and minimizing trade-offs in each UWS process, such as changing water resources, leakage management strategies (e.g., active or passive leakage control), technologies in water and wastewater treatment, indoor water heating energy types, and water reuse [71–73]. In this study, twelve scenarios were developed and analyzed, as shown in Table 4, to identify the most appropriate urban water loss management strategy and quantify resource movement between water and energy sectors according to various energy intensities and water loss status. According to the energy intensities of UWS, cities were classified into three groups (i.e., low-, medium-, and high-intensity cities). A city with low energy intensity is a city that requires less energy to service a unit of water during UWS compared to a city with high energy intensity. The variables and energy intensity values were set based on previous literature, as introduced in Section 2.2, and the details are presented in Table 5. Cities were classified into four types of leakage status, as shown in Table 5, including Good Condition, High NRW & Low AL, High NRW & Medium AL, and High NRW & High AL. In the case of Good Condition, water loss is well-managed, and no further improvement is needed. The High NRW & Low AL case refers to high water loss and RL caused by pipe bursts as the main problem. In the High NRW & High AL case, plenty of AL is occurring due to significant inefficiencies in the recording, archiving, and accounting operation used to track water volume in the water utility. In the High NRW & Medium case, the amount of AL is between those of AL in High NRW & Low AL and High NRW & High AL.

**Table 4.** Simulation scenarios according to water loss conditions and energy intensity.

| Energy Intensity \ Water Losses | Good Condition | High NRW & Low AL Ratio | High NRW & Medium AL Ratio | High NRW & High AL Ratio |
|---|---|---|---|---|
| Low EI | S 1-1 | S 1-2 | S 1-3 | S 1-4 |
| Medium EI | S 2-1 | S 2-2 | S 2-3 | S 2-4 |
| High EI | S 3-1 | S 3-2 | S 3-3 | S 3-4 |

**Table 5.** The value of energy intensity in low-, medium-, and high-intensity cases.

| Urban Water System | Parameter | Low EI | Medium EI | High EI |
|---|---|---|---|---|
| Intake | Energy intensity | 0.0027 | 0.0027 | 0.0027 |
| | Groundwater level | 20 | 40 | 60 |
| | Groundwater ratio | 0.1 | 0.5 | 0.9 |
| Conveyance and transmission | Energy intensity | 0.2 | 2.1 | 4 |
| Water treatment | Energy intensity | 0.2 | 0.6 | 1 |
| Water distribution | Energy intensity | 0.2 | 0.5 | 0.8 |
| Use | Energy intensity | 50 | 50 | 50 |
| | Hot water usage ratio | 0.01 | 0.02 | 0.03 |
| Sewage collection | Energy intensity | 0 | 0 | 0 |
| | Sewage collection ratio | 0.9 | 0.9 | 0.9 |
| Wastewater treatment | Energy intensity | 0.3 | 0.65 | 1 |
| | WW treatment ratio | 0.9 | 0.9 | 0.9 |
| Discharge | Energy intensity | 0.02 | 0.02 | 0.02 |

The average NRW ratio varies from country to country. As of 2017, the NRW ratio worldwide had an average value of 0.31 [128]. Kuwait, Singapore, and the Netherlands had very low NRW ratio values of 0.03, 0.05, and 0.05, respectively. In contrast, Bulgaria, Turkey, and Venezuela had high NRW ratio values of 0.61, 0.61, and 0.63, respectively. Water utilities derive the targets of water loss control programs by contemplating political considerations, water conservation issues, customer supply security, economic considerations, and/or regulatory requirements [7]. The economic level of leakage impact is the break-even point beyond which the effort to control the losses costs more than the value of the recovery, and is mainly used as an indicator for the target [129]. In general, water utilities conduct water loss control programs to maintain NRW in the range of 0.15 to 0.20, and they should also aim for no more than 4–6% of AL [130]. Water loss management is a painstaking process, and a realistic time scale should be chosen for leakage targets. Short-term (e.g., managing the backlog of leakage), medium-term (e.g., installing DMAs and pressure management), and long-term (e.g., replacing mains and service connections) countermeasures should be included in this program. Therefore, water loss control programs generally require a trajectory of more than five years. In the simulations for this study, each high NRW case (i.e., High NRW & Low AL, High NRW & Medium AL, High NRW & High AL) was designed to reach Good Condition after five years. In Good Condition, the NRW ratio is 0.2, of which 5% is AL, and 95% is RL. In high NRW cases, the NRW ratio was set to 0.5, and the proportion of AL in low AL, medium AL, and high AL in NRW was assumed to be 10%, 30%, and 50%, respectively. In developed countries, RL usually represents the most critical component of water loss, as in the case of High NRW & Low AL. However, in developing and emerging countries, AL due to illegal connections, metering inaccuracy, and accounting errors may be of major significance to water utilities, as in cases of High NRW & Medium AL or High NRW & High AL. The final values and reduction rates for this purpose are presented in Table 6.

**Table 6.** The values of apparent losses and real losses in each scenario.

| | Apparent Loss (NRW a) | | | | Real Loss (NRW r) | | | |
|---|---|---|---|---|---|---|---|---|
| | Initial Value | Final Value | Reduction Rate [L/Month] | Month of Final Value | Initial Value | Final Value | Reduction Rate [L/Month] | Month of Final Value |
| Good condition | 0.01 | 0.01 | 0 | - | 0.19 | 0.19 | 0 | - |
| High NRW & Low AL ratio | 0.05 | 0.01 | 0.00067 | 60 | 0.45 | 0.19 | 0.00433 | 60 |
| High NRW & Medium AL ratio | 0.15 | 0.01 | 0.00233 | 60 | 0.35 | 0.19 | 0.00267 | 60 |
| High NRW & High AL ratio | 0.25 | 0.01 | 0.004 | 60 | 0.25 | 0.19 | 0.001 | 60 |

The developed WE Nexus model contains several limitations. It is challenging to derive detailed implementation through the developed model; as such, the model did not consider the entire process of UWS and was simplified for trend analysis. In addition, the developed model was not applied to actual urban cases and was constructed using data presented in the literature. Despite these limitations, the developed WE Nexus model enables the quantification of synergies and trade-offs between sectors through connection analysis for each sector, thereby enabling a water loss management strategy from the Nexus perspective.

## 3. Results

### 3.1. The Effect of Leakage Status on the WE Nexus

Based on the twelve scenarios introduced, changes in water footprint, total energy use, and carbon footprint for ten years are shown in Tables 7–9. Since the water loss conditions for each scenario are the same after five years, if the urban intensity is the same, the values of water footprint, total energy use, and carbon footprint remain the same after five years. In cities with the same energy intensity in UWS, water footprint, total energy use, and carbon footprint resulted in order of High NRW & Low AL, High NRW & Medium AL, High NRW & High AL, and Good Condition. This indicates that the lower the AL in the same NRW ratio, the lower the loss WE Nexus perspective. In other words, although AL can adversely impact water utilities' profit, it does not require additional resources in the urban WE Nexus perspectives. However, RL increases water utilities' production costs and equates to the resource loss in urban WE Nexus. In this context, reducing RL is essential to sustainably supply water in urban areas from the perspective of the WE Nexus.

**Table 7.** Change of water footprint [$10^6$ m$^3$] by scenario.

| Time [Month] Scenario | 0 | 1 | 13 | 25 | 37 | 49 | 61 | 73 | 85 | 97 | 109 | 120 |
|---|---|---|---|---|---|---|---|---|---|---|---|---|
| S 1-1 | 1.13951 | 1.14065 | 1.15441 | 1.16834 | 1.18244 | 1.19680 | 1.21115 | 1.22576 | 1.24055 | 1.25552 | 1.27067 | 1.28472 |
| S 1-2 | 1.67129 | 1.66 | 1.53721 | 1.43406 | 1.34622 | 1.27869 | 1.21115 | 1.22576 | 1.24055 | 1.25552 | 1.27067 | 1.28472 |
| S 1-3 | 1.41631 | 1.412 | 1.36296 | 1.3185 | 1.27804 | 1.24460 | 1.21115 | 1.22576 | 1.24055 | 1.25552 | 1.27067 | 1.28472 |
| S 1-4 | 1.22933 | 1.22895 | 1.22448 | 1.22033 | 1.21649 | 1.21382 | 1.21115 | 1.22576 | 1.24055 | 1.25552 | 1.27067 | 1.28472 |
| S 2-1 | 1.21303 | 1.21424 | 1.2289 | 1.24372 | 1.25873 | 1.27401 | 1.28929 | 1.30485 | 1.32059 | 1.33653 | 1.35265 | 1.36761 |
| S 2-2 | 1.77259 | 1.76072 | 1.63169 | 1.52332 | 1.43107 | 1.36018 | 1.28929 | 1.30485 | 1.32059 | 1.33653 | 1.35265 | 1.36761 |
| S 2-3 | 1.5042 | 1.49968 | 1.44826 | 1.40167 | 1.35929 | 1.32429 | 1.28929 | 1.30485 | 1.32059 | 1.33653 | 1.35265 | 1.36761 |
| S 2-4 | 1.30738 | 1.30699 | 1.30249 | 1.29833 | 1.2945 | 1.29190 | 1.28929 | 1.30485 | 1.32059 | 1.33653 | 1.35265 | 1.36761 |
| S 3-1 | 1.29196 | 1.29325 | 1.30885 | 1.32465 | 1.34063 | 1.35691 | 1.37318 | 1.38975 | 1.40651 | 1.42349 | 1.44066 | 1.45659 |
| S 3-2 | 1.88185 | 1.86935 | 1.73349 | 1.6194 | 1.52231 | 1.44775 | 1.37318 | 1.38975 | 1.40652 | 1.42349 | 1.44066 | 1.45659 |
| S 3-3 | 1.59882 | 1.59407 | 1.54004 | 1.49111 | 1.44661 | 1.40990 | 1.37318 | 1.38975 | 1.40651 | 1.42349 | 1.44066 | 1.45659 |
| S 3-4 | 1.39127 | 1.39087 | 1.38632 | 1.38213 | 1.37827 | 1.37573 | 1.37318 | 1.38975 | 1.40651 | 1.42349 | 1.44066 | 1.45659 |

**Table 8.** Change of total energy use [$10^6$ kWh] by scenario.

| Time [Month] Scenario | 0 | 1 | 13 | 25 | 37 | 49 | 61 | 73 | 85 | 97 | 109 | 120 |
|---|---|---|---|---|---|---|---|---|---|---|---|---|
| S 1-1 | 1.40075 | 1.40215 | 1.41907 | 1.43619 | 1.45352 | 1.47117 | 1.48881 | 1.50677 | 1.52495 | 1.54335 | 1.56198 | 1.57924 |
| S 1-2 | 1.72189 | 1.71578 | 1.65024 | 1.59665 | 1.55243 | 1.52062 | 1.48881 | 1.50677 | 1.52495 | 1.54335 | 1.56198 | 1.57924 |
| S 1-3 | 1.56322 | 1.56142 | 1.54148 | 1.52433 | 1.50963 | 1.49922 | 1.48881 | 1.50677 | 1.52495 | 1.54335 | 1.56198 | 1.57924 |
| S 1-4 | 1.44687 | 1.44749 | 1.45505 | 1.46289 | 1.471 | 1.47991 | 1.48881 | 1.50677 | 1.52495 | 1.54335 | 1.56198 | 1.57924 |
| S 2-1 | 5.07679 | 5.08186 | 5.14318 | 5.20524 | 5.26805 | 5.33200 | 5.39595 | 5.46105 | 5.52695 | 5.59364 | 5.66113 | 5.72372 |
| S 2-2 | 6.78688 | 6.75198 | 6.37419 | 6.05973 | 5.79474 | 5.59535 | 5.39595 | 5.46105 | 5.52695 | 5.59364 | 5.66113 | 5.72372 |
| S 2-3 | 5.95747 | 5.9452 | 5.80669 | 5.68298 | 5.5722 | 5.48408 | 5.39595 | 5.46105 | 5.52695 | 5.59364 | 5.66113 | 5.72372 |
| S 2-4 | 5.34924 | 5.34969 | 5.35571 | 5.36294 | 5.37133 | 5.38364 | 5.39595 | 5.46105 | 5.52695 | 5.59364 | 5.66113 | 5.72372 |
| S 3-1 | 9.02305 | 9.03207 | 9.14105 | 9.25135 | 9.36298 | 9.47664 | 9.59029 | 9.70601 | 9.82313 | 9.94166 | 10.0616 | 10.1728 |
| S 3-2 | 12.2498 | 12.1834 | 11.4639 | 10.8637 | 10.3568 | 9.97355 | 9.5903 | 9.70602 | 9.82313 | 9.94166 | 10.0616 | 10.1729 |
| S 3-3 | 10.6885 | 10.6647 | 10.3958 | 10.1548 | 9.93814 | 9.76422 | 9.59029 | 9.70601 | 9.82313 | 9.94166 | 10.0616 | 10.1728 |
| S 3-4 | 9.54345 | 9.54363 | 9.547 | 9.55256 | 9.56025 | 9.57527 | 9.59029 | 9.70601 | 9.82313 | 9.94166 | 10.0616 | 10.1728 |

**Table 9.** Change of carbon footprint [$10^6$ kgCO$_2$eq] by scenario.

| Time [Month] Scenario | 0 | 1 | 13 | 25 | 37 | 49 | 61 | 73 | 85 | 97 | 109 | 120 |
|---|---|---|---|---|---|---|---|---|---|---|---|---|
| S 1-1 | 0.38858 | 0.38897 | 0.39367 | 0.39842 | 0.40322 | 0.40812 | 0.41301 | 0.41800 | 0.42304 | 0.42815 | 0.43331 | 0.43810 |
| S 1-2 | 0.47925 | 0.47752 | 0.45893 | 0.44372 | 0.43115 | 0.42208 | 0.41301 | 0.41800 | 0.42304 | 0.42815 | 0.43331 | 0.43810 |
| S 1-3 | 0.43432 | 0.43381 | 0.42812 | 0.42323 | 0.41902 | 0.41602 | 0.41301 | 0.41800 | 0.42304 | 0.42815 | 0.43331 | 0.43810 |
| S 1-4 | 0.40137 | 0.40154 | 0.40364 | 0.40582 | 0.40807 | 0.41054 | 0.41301 | 0.41800 | 0.42304 | 0.42815 | 0.43331 | 0.43810 |
| S 2-1 | 1.3076 | 1.30891 | 1.3247 | 1.34068 | 1.35686 | 1.37333 | 1.3898 | 1.40657 | 1.42354 | 1.44072 | 1.45811 | 1.47423 |
| S 2-2 | 1.7455 | 1.73657 | 1.63992 | 1.55949 | 1.49173 | 1.44077 | 1.3898 | 1.40657 | 1.42354 | 1.44072 | 1.45811 | 1.47423 |
| S 2-3 | 1.53289 | 1.52976 | 1.49443 | 1.46289 | 1.43466 | 1.41223 | 1.3898 | 1.40657 | 1.42354 | 1.44072 | 1.45811 | 1.47423 |
| S 2-4 | 1.37697 | 1.3771 | 1.37881 | 1.38084 | 1.38316 | 1.38648 | 1.3898 | 1.40657 | 1.42354 | 1.44072 | 1.45811 | 1.47423 |
| S 3-1 | 2.29416 | 2.29646 | 2.32417 | 2.35221 | 2.38059 | 2.40949 | 2.43839 | 2.46781 | 2.49759 | 2.52773 | 2.55823 | 2.58651 |
| S 3-2 | 3.11124 | 3.09443 | 2.91234 | 2.76048 | 2.63225 | 2.53532 | 2.43839 | 2.46781 | 2.49759 | 2.52773 | 2.55823 | 2.58651 |
| S 3-3 | 2.71563 | 2.70962 | 2.6417 | 2.58084 | 2.52615 | 2.48227 | 2.43839 | 2.46781 | 2.49759 | 2.52773 | 2.55823 | 2.58651 |
| S 3-4 | 2.42552 | 2.42558 | 2.42663 | 2.42824 | 2.43039 | 2.43439 | 2.43839 | 2.46781 | 2.49759 | 2.52773 | 2.55823 | 2.58651 |

For high-water-energy-intensity cities, the initial water footprint from scenarios of Good Condition (S 3-1), High NRW & Low AL (S 3-2), High NRW & Medium AL (S 3-3), and High NRW & High AL (S 3-3) showed $1.292 \times 10^6$ m$^3$, $1.883 \times 10^6$ m$^3$, $1.598 \times 10^6$ m$^3$, and $1.391 \times 10^6$ m$^3$, respectively, as shown in Figure 3. S 3-2, S 3-3, and S 3-4 showed high values of 45.7%, 23.7%, and 7.7%, respectively, as compared to S 3-1. As demonstrated in Figure 4, total energy uses of the same scenarios are $9.023 \times 10^6$ kWh, $12.250 \times 10^6$ kWh, $10.689 \times 10^6$ kWh, and $9.543 \times 10^6$ kWh, respectively, and were 35.8%, 18.5%, and 5.7% higher than the Good Condition scenario, respectively. The carbon footprint for the same scenarios was calculated as $2.294 \times 10^6$ kgCO$_2$eq, $3.11 \times 10^6$ kgCO$_2$eq, $2.716 \times 10^6$ kgCO$_2$eq, and $2.426 \times 10^6$ kgCO$_2$eq, respectively, as shown in Figure 5. These values of S 3-2, S 3-3, and S 3-4 were 35.6%, 18.4%, and 5.8% higher than the Good Condition scenario, respectively. When water utilities establish a water loss control program, they generally focus on AL because it is low-hanging fruit. However, these results indicated that reducing AL does not contribute much from the WE Nexus perspective, and reducing RL is essential.

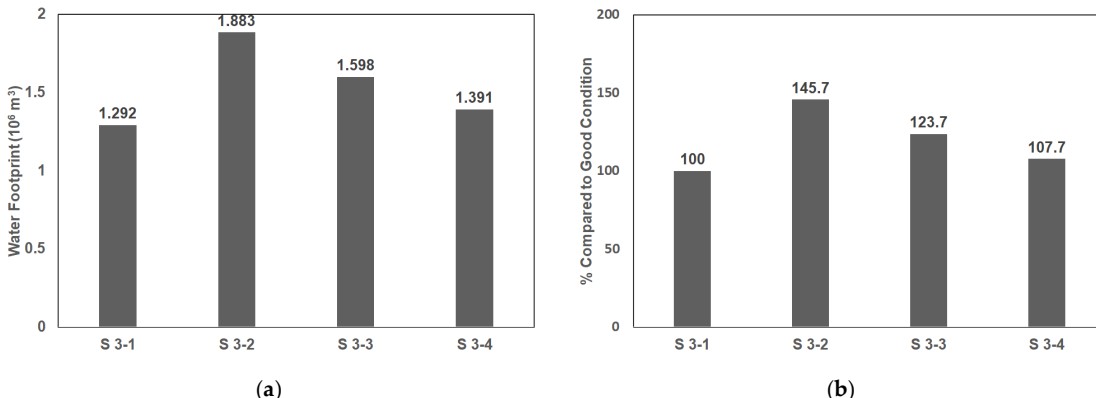

**Figure 3.** Water footprint according to water loss conditions in the High Energy Intensity scenario: (**a**) Total amount; (**b**) Ratio.

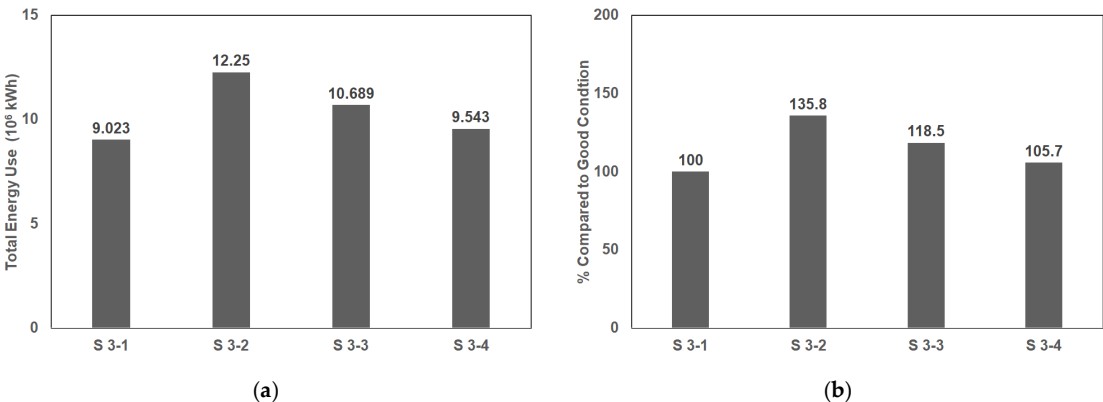

**Figure 4.** Total energy use according to water loss conditions in the High Energy Intensity scenario: (**a**) Total amount; (**b**) Ratio.

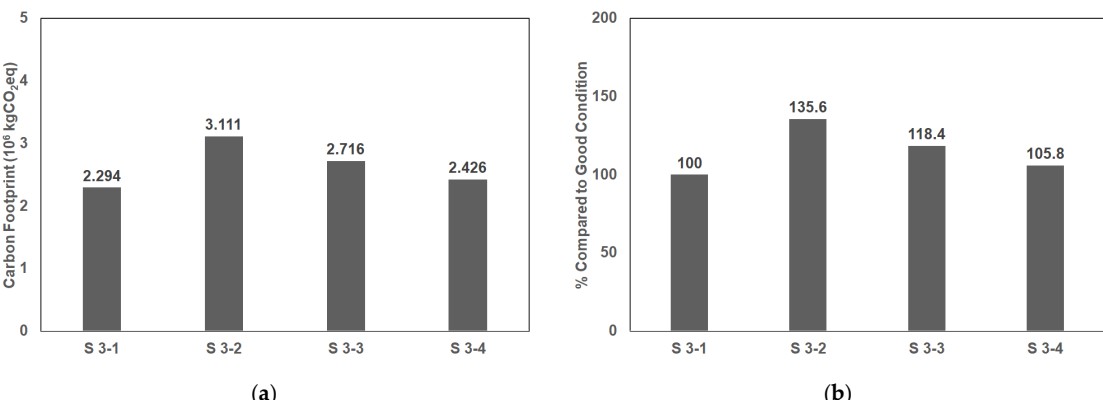

**Figure 5.** Carbon footprint according to water loss conditions in the High Energy Intensity scenario: (**a**) Total amount; (**b**) Ratio.

*3.2. The Effect of Energy Intensity on the WE Nexus*

In cities with the same water leakage status, water footprint, total energy use, and carbon footprint resulted in the order of cities with high energy intensity, medium energy intensity, and low energy intensity. Under the Good Condition, the initial water footprints in low (S 1-1), medium (S 2-1), and high energy intensity (S 3-1) cases were $1.139 \times 10^6$ m$^3$, $1.213 \times 10^6$ m$^3$, and $1.598 \times 10^6$ m$^3$, respectively as shown in Figure 6. Water footprints in cities with medium and high energy intensity were found to be 6.5% and 40.3% higher than in the low energy intensity case. Total energy usage of S 1-1, S 2-1, and S 3-1 were $1.401 \times 10^6$ kWh, $5.077 \times 10^6$ kWh, and $9.023 \times 10^6$ kWh, respectively as shown in Figure 7, and cities with medium energy intensity and high energy intensity required 262.4% and 644.0% more than the city with low energy intensity, respectively. The carbon footprints of S 1-1, S 2-1, and S 3-1 were $0.389 \times 10^6$ kgCO$_2$eq, $1.308 \times 10^6$ kgCO$_2$eq, and $2.294 \times 10^6$ kgCO$_2$eq for each scenario as shown in Figure 8, and based on the low energy intensity city, carbon footprint growth rates in medium and high energy intensity cities were 236.3% and 589.8%, respectively. The water footprint assessment revealed an increasing trend that as cities' energy intensity increased, the total energy intensity rapidly increased compared to the water footprint. As the carbon footprint is directly related to total energy intensity, the value for the carbon footprint increased proportionally with total energy intensity. Water utilities do not typically consider energy intensity when establishing water loss control strategies. However, the energy intensity of UWS should be considered a crucial factor when establishing strategies from a WE Nexus perspective.

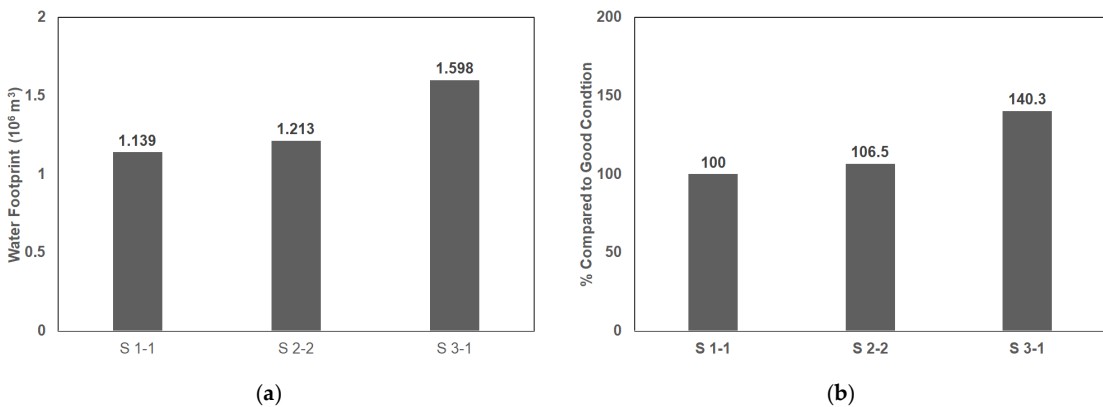

(a)              (b)

**Figure 6.** Water footprint according to energy intensity of UWS in the Good Condition scenario: (**a**) Total amount; (**b**) Ratio.

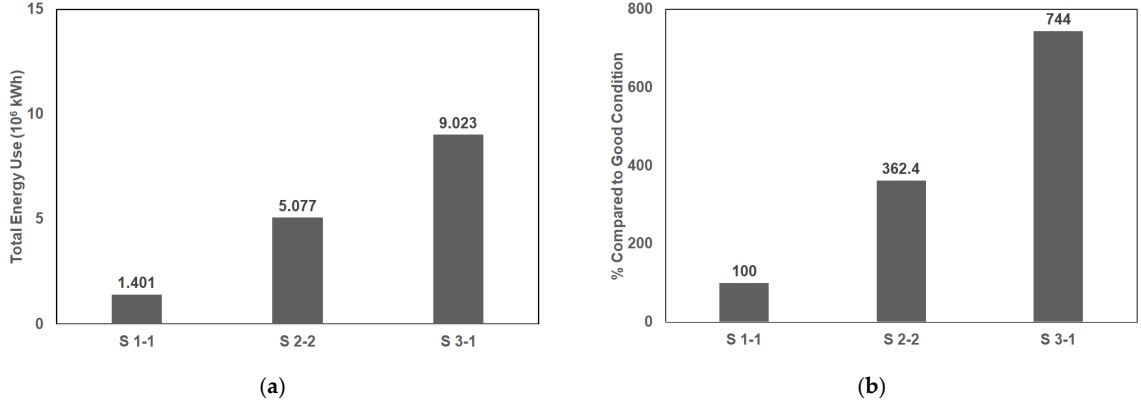

(a)              (b)

**Figure 7.** Total energy use according to energy intensity of UWS in the Good Condition scenario: (**a**) Total amount; (**b**) Ratio.

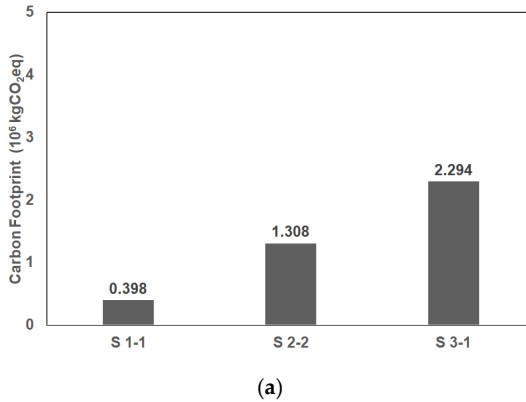 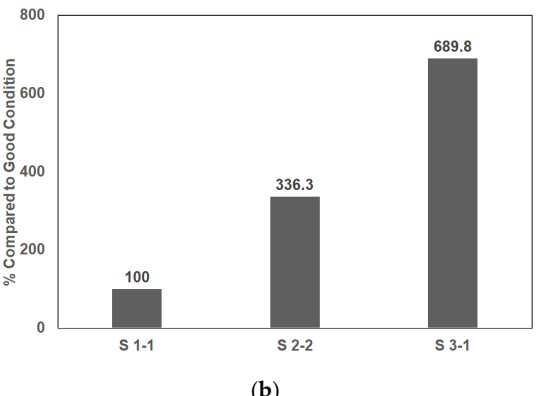

(**a**)  (**b**)

**Figure 8.** Carbon footprint according to energy intensity of UWS in the Good Condition scenario: (**a**) Total amount; (**b**) Ratio.

*3.3. Water Footprint, Total Energy Use, and Carbon Footprint in Each UWS Process*

Scenario S 3-1 was analyzed to investigate resource use and transfer in UWS, including abstraction, conveyance, transmission, water treatment, end-uses, wastewater collection, wastewater treatment, and disposal. In the initial stage prior to the implementation of the water loss management program in the S 3-1 scenario (t = 0 month), the total amounts and ratio of water footprint, total energy use, and carbon footprint during the process of UWS are presented in Figures 9 and 10. During these processes, the total water footprint was 1,881,851 m³ and the water footprints of abstraction, conveyance, water treatment, distribution, end-use, wastewater collection, wastewater treatment, and wastewater disposal were 1,641,147 m³ (87.21%), 131,170 m³ (6.97%), 33,120 m³ (1.76%), 32,793 m³ (1.74%), 27,054 m³ (1.44%), 0 m³ (0%), 16,245 m³ (0.86%), and 322 m³ (0.02%), respectively. The abstraction directly used 87.21% of the total water footprint, indicating that the majority of the water footprint occurred in the abstraction process. In the rest of UWS, except for abstraction, water footprints were generated by the energy or chemicals used in each process, and water footprints with energy and chemicals accounted for 13.02% and 0.03%, respectively. Total energy uses of abstraction, conveyance, water treatment, distribution, end-use, wastewater collection, wastewater treatment, and wastewater disposal processes were 238,582 kWh (1.95%), 6,545,450 kWh (53.43%), 1,652,724 kWh (13.49%), 1,636,360 kWh (13.36%), 1,350,000 kWh (11.02%), 0 kWh (0%), 810,662 kWh (6.62%), and 16,053 kWh (0.13%), respectively. Under the processes of UWS, a significant portion of energy was used during conveyance, followed by water treatment, distribution, and end-use. Carbon footprints in water abstraction, conveyance, water treatment, distribution, end-use, wastewater collection, wastewater treatment, and wastewater disposal were calculated as 59,765 kgCO₂eq (1.92%), 1,639,628 kgCO₂eq (52.70%), 442,634 kgCO₂eq (14.23%), 409,906 kgCO₂eq (13.18%), 338,174 kgCO₂eq (10.87%), 0 kgCO₂eq (0%), 217,112 kgCO₂eq (6.98%), and 4021 kgCO₂eq (0.13%), respectively. Since carbon emission occurred from energy generation, the results of carbon footprint are similar to the results of total energy use. In the S 3-1 scenario, the total amounts and ratios of water footprint, total energy use, and carbon footprint in UWS when approaching the Good Condition (t = 61 month) are presented in Figures 11 and 12. Water footprint, total energy use, and carbon footprint were reduced by 27%, 22%, and 22%, respectively, by applying the effective water loss management program. In the entire UWS, water footprint, total energy intensity, and carbon footprint decreased significantly in abstraction, conveyance, water treatment, and distribution processes, while slightly increasing in end-use, wastewater treatment, and wastewater disposal processes.

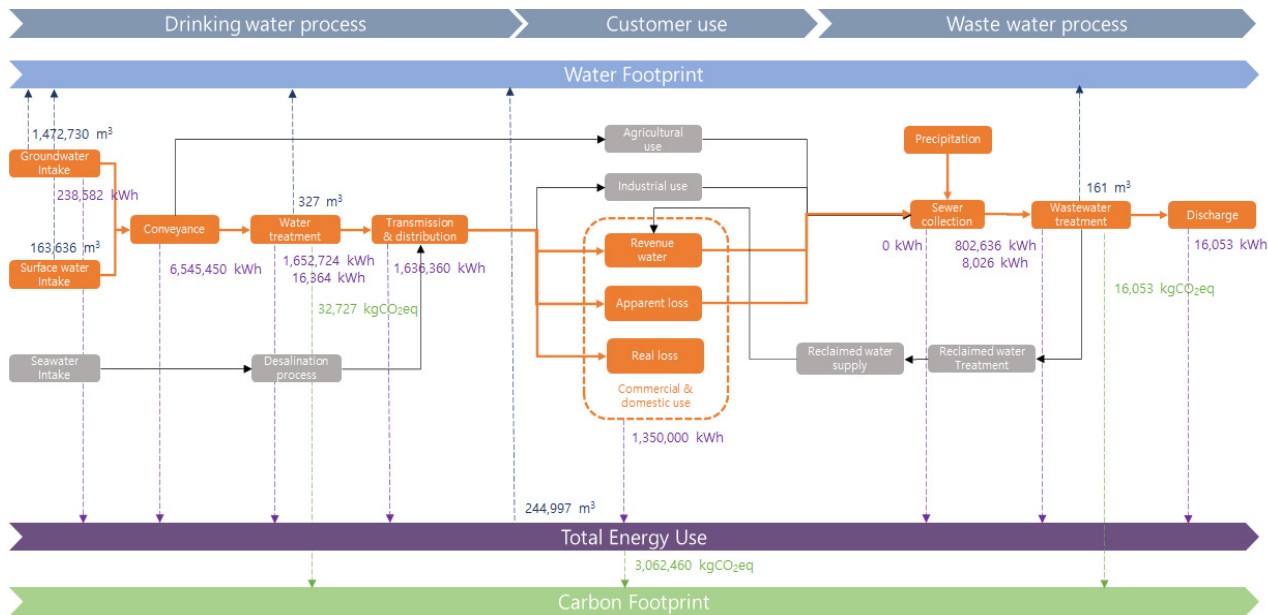

**Figure 9.** The amount of water footprint, total energy use, and carbon footprint in the process of UWS at the initial stage (t = 0 month) of the S 3-1 scenario.

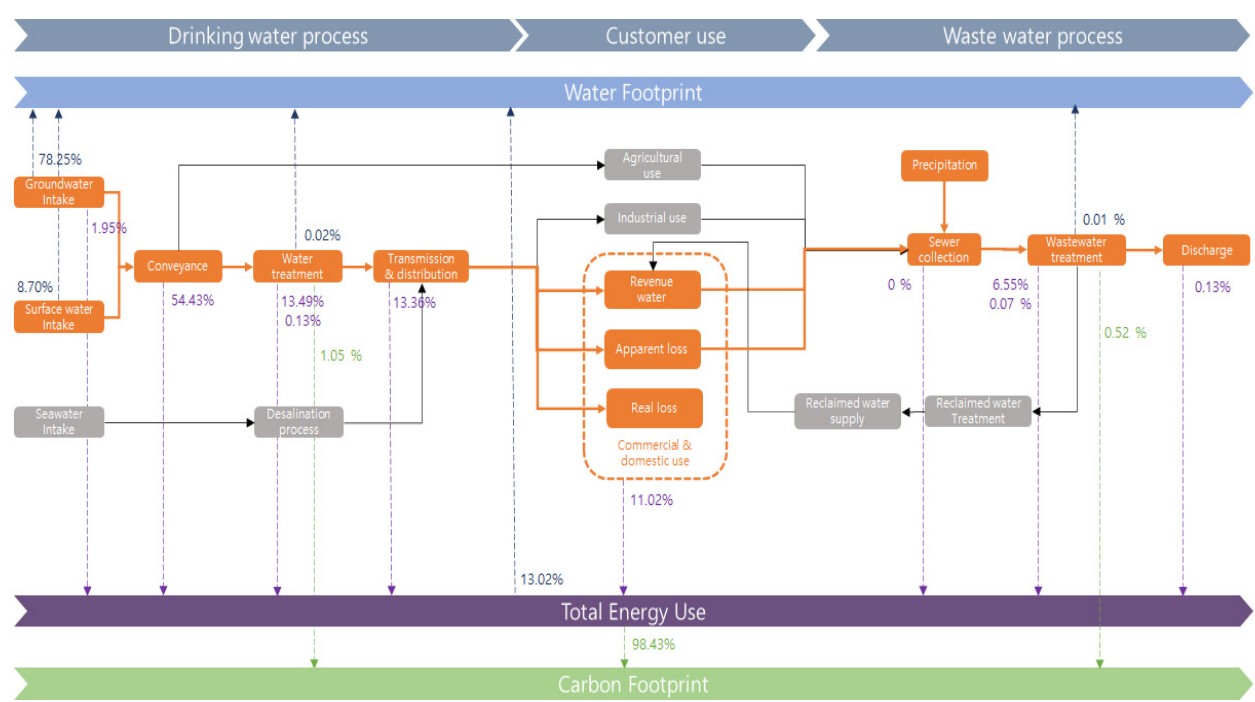

**Figure 10.** The ratio of water footprint, total energy use, and carbon footprint in the process of UWS at the initial stage (t = 0 month) of the S 3-1 scenario.

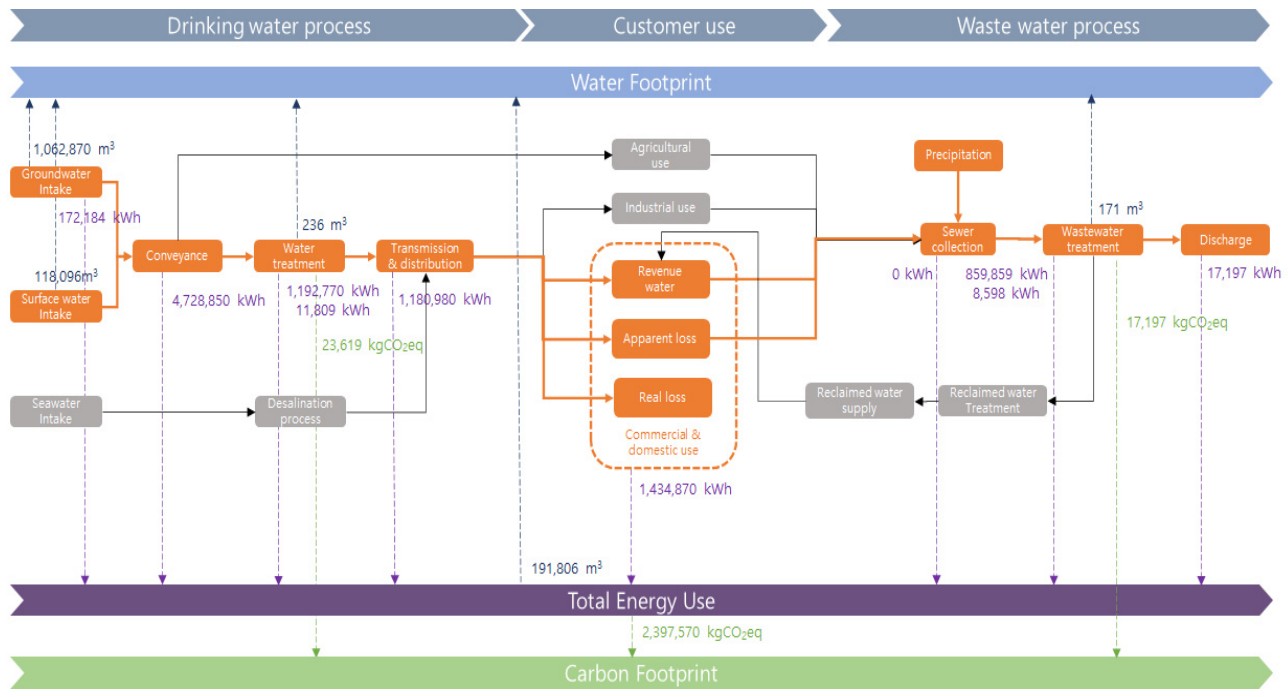

**Figure 11.** The amount of water footprint, total energy use, and carbon footprint in the process of UWS at the stabilized stage (t = 61 months) of the S 3-1 scenario.

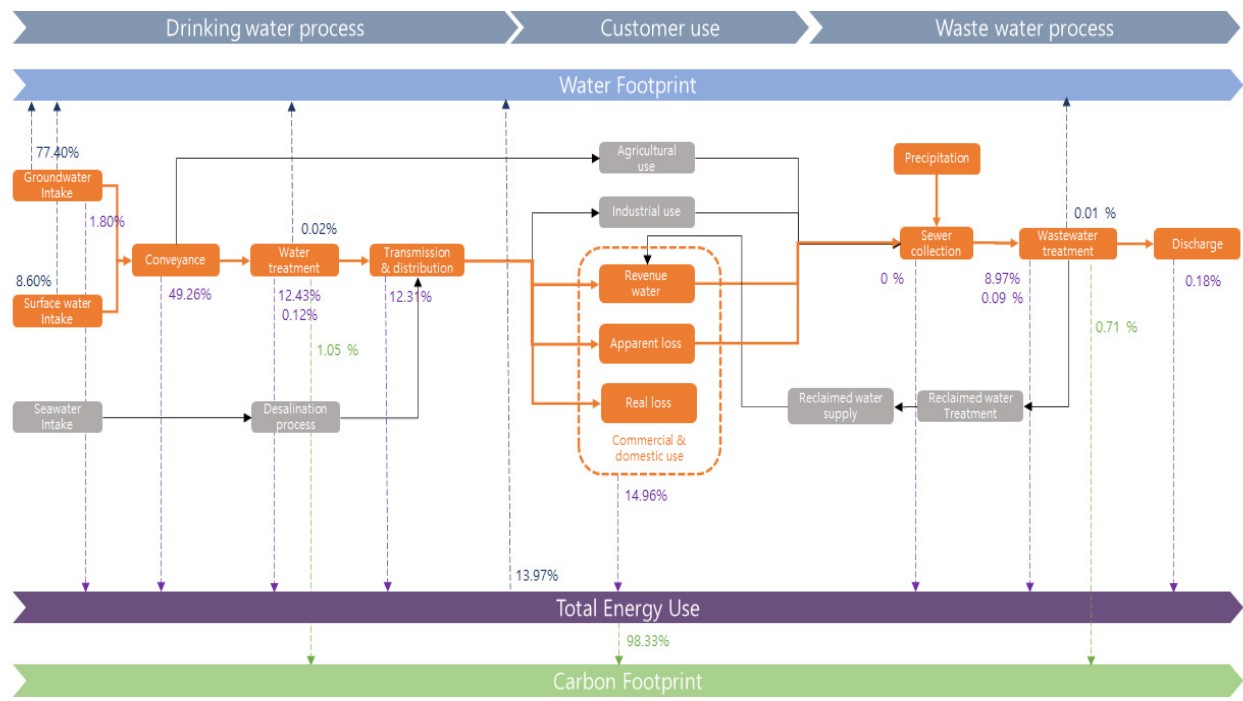

**Figure 12.** The ratio of water footprint, total energy use, and carbon footprint in the process of UWS at the stabilized stage (t = 61 months) of the S 3-1 scenario.

## 4. Discussion

Strategies for water loss management from the WE Nexus point of view were suggested by analyzing twelve scenarios considering three urban energy intensity statuses and four water loss statuses.

Reducing AL was prioritized in the conventional economic-oriented water loss control program; however, handling RL should be considered an essential aspect from the WE Nexus perspective. Water utilities establish water loss control programs through analyzing present status (i.e., preliminary system diagnosis and water audit) to quantify the volume of water loss and identify its location. Then appropriate interventions are selected and detailed action plans are developed considering the economic level of leakage and target-setting guidelines. In the intervention selection process, water utilities first focus on the AL from the financial perspective. However, this WE Nexus study confirmed that RL is the main cause of resource waste. Therefore, reducing RL should be considered an essential direction in water loss control programs.

The energy intensity of each process of UWS was demonstrated to have a significant impact on resource consumption and transfer. Therefore, the energy intensity of UWS should also be considered a crucial factor for advanced analyses. Previous studies have not focused on energy intensity in UWS, and water and energy systems are often operated and managed separately; however, this study confirmed that a difference in the energy intensity of UWS yields a difference of about 40.3%, 644%, and 590% in water footprint, total energy use, and carbon footprint, respectively.

The WE Nexus model quantified the consumption and movement of resources in each process of UWS and identified central and vulnerable processes. Abstraction from a water perspective and conveyance from an energy perspective were revealed as the most influential processes in the WE Nexus. However, these results are expected to differ based on UWS conditions, such as water source quality, topography, applied technology, and the efficiency of water treatment and delivery systems.

The WE Nexus model has several limitations (e.g., a water-driven Nexus, one-way impact analysis, not considering all UWS processes, not including detailed causality loops, and not applying the model in actual city case data); however, the comprehensive WE Nexus model for UWS using SD enabled the quantitative calculation of the amount of use and movement of resources between water and energy sectors. Therefore, discussions regarding the development and subsequent monitoring of the SDGs are possible through a Nexus study. In addition, the application of the model contributes to the establishment of sustainable, systematic, and feasible water loss management strategies from the Nexus perspective.

## 5. Conclusions

New management approaches are needed to ensure water and energy security. The concept of Nexus in UWM has been identified as a beneficial approach by governments, industries, and researchers in the past years as it can quantify and assess the real-world interlinkages between sectors to reach ultimate goals. To adopt a holistic Nexus view, the following significant issues should be addressed.

Site-specific and individual Nexus research requires diverse data and information, with different benefits and limitations. Such models will only be valid in particular conditions and accepted under certain constraints. Therefore, a comprehensive, versatile, practical, and widely accepted Nexus framework and methodologies should be developed to assess and coordinate the actions of diverse stakeholders and facilitate decision-making although it is difficult to correspond to every unique Nexus case.

Institutional collaboration and coordination tools are essential components for implementing Nexus research results into actual policies and practices. Therefore, Nexus models should focus on physical connections and policy interventions as well as institutional governance to overcome non-uniform regulations, isolated decisions, and policy

planning in each sector. After identifying institutional barriers, such as vertically structured government departments and sector-oriented policies through Nexus research, cross-sectoral policy coherence can be achieved.

Although the Nexus concept has been popular in academia, business, agencies, and governments, research has not suggested how to shift the Nexus concept to advance practices on the ground. While bountiful literature aims to provide insight, less research is designed to support governance and implementation. Therefore, research on implementing, rather than simply understanding the Nexus perspective should be a central focus of future research. For Nexus research to receive continuous attention, it is necessary to demonstrate practical results in the implementation stage.

**Author Contributions:** Conceptualization, B.S. and E.S.; methodology, S.H.C. and E.S.; software, S.H.C. and E.S.; formal analysis, S.H.C. and E.S.; writing—original draft preparation, S.H.C.; writing—review and editing, E.S.; visualization, S.H.C. and E.S.; supervision, B.S. and E.S. All authors have read and agreed to the published version of the manuscript.

**Funding:** This research received no external funding.

**Institutional Review Board Statement:** Not applicable.

**Informed Consent Statement:** Not applicable.

**Data Availability Statement:** No new data were created or analyzed in this study. Data sharing is not applicable to this article.

**Conflicts of Interest:** The authors declare no conflict of interest.

## Abbreviation

List of Acronyms:

| | |
|---|---|
| AL | Apparent Loss |
| CLEW3 | Climate, Land-use, Energy and Water 3 |
| GHG | Greenhouse Gas |
| DMA | District Metered Area |
| GAEZ | Global Agro-Ecological Zones |
| IWRM | Integrated Water Resources Management |
| LEAP | Low Emissions Analysis Platform |
| MESSAGE | Model for Energy Supply Strategy Alternatives and their General Environmental Impacts |
| MuSIASEM | Multi-Scale Integrated Analysis of Societal and Ecosystem Metabolism |
| NRW | Non-revenue Water |
| RL | Real Loss |
| SD | System Dynamics |
| SDGs | Sustainable Development Goals |
| UWC | Urban Water Cycle |
| UWS | Urban Water System |
| WEAP | Water Evaluation and Planning System |
| WE | Water-Energy |

## Appendix A

**Table A1.** Variables and Equations in UWS WE Nexus.

| No. | Variables | Equations | Unit |
|---|---|---|---|
| 1 | Apparent loss | NRW a ratio/(1-NRW r ratio) * "Residential & Commercial water use" | $m^3$ |
| 2 | Apparent loss ratio | Apparent loss/(Apparent loss + Real loss) | - |
| 3 | CF coef for te | 0.25 | $kgCO_2e/kWh$ |
| 4 | CF coef for wtc | 1e-06 | $kgCO_2e/mg$ |
| 5 | CF coef for wwtc | 1e-06 | $kgCO_2e/mg$ |
| 6 | CF of total energy | CF coef for te * Total Energy for UWS | $kgCO_2e$ |
| 7 | CF of wastewater treatment chemical | CF coef for wwtc * Major chemicals for wastewater treatment | $kgCO_2e$ |
| 8 | CF of water treatment chemical | CF coef for wtc * Major chemicals for water treatment | $kgCO_2e$ |

| 9 | Conveyance | Water treatment | $m^3$ |
|---|---|---|---|
| 10 | Conveyance energy | Energy intensity c * Conveyance | kWh |
| 11 | Discharge | Wastewater treatment | $m^3$ |
| 12 | Discharge energy | Energy intensity d * Discharge | kWh |
| 13 | EE coef for wtc | 5e-07 | kWh/mg |
| 14 | EE coef for wwtc | 5e-07 | kWh/mg |
| 15 | Embodied energy of wastewater treatment chemical | EE coef for wwtc * Major chemicals for wastewater treatment | kWh |
| 16 | Embodied energy of water treatment chemical | EE coef for wtc * Major chemicals for water treatment | kWh |
| 17 | End use energy | Water heating energy | kWh |
| 18 | Energy intensity c | 0.2, 2.1, 4 | kWh/$m^3$ |
| 19 | Energy intensity d | 0.02 | kWh/$m^3$ |
| 20 | Energy intensity gi | 0.0027 | kWh/$m^3$/m |
| 21 | Energy intensity si | 0 | kWh/$m^3$ |
| 22 | Energy intensity td | 0.2, 0.5, 0.8 | kWh/$m^3$ |
| 23 | Energy intensity wh | 50 | kWh/$m^3$ |
| 24 | Energy intensity wt | 0.2, 0.6, 1 | kWh/$m^3$ |
| 25 | Energy intensity wwc | 0 | kWh/$m^3$ |
| 26 | Energy intensity wwt | 0.3, 0.65, 1 | kWh/$m^3$ |
| 27 | Groundwater intake | Groundwater ratio * Conveyance | $m^3$ |
| 28 | Groundwater level | 20, 40, 60 | m |
| 29 | Groundwater ratio | 0.1, 0.5, 0.9 | - |
| 30 | Intake energy | (Energy intensity gi*Groundwater level*Groundwater intake) + (Energy intensity si *Surface water intake) | kWh |
| 31 | lpcd | 100 | liter/Person/day |
| 32 | lpcd unit conversion | 0.03 | (day*$m^3$)/liter |
| 33 | Major chemicals for wastewater treatment | Wastewater chemical dosage * Wastewater treatment | mg |
| 34 | Major chemicals for water treatment | Water chemical dosage * Water treatment | mg |
| 35 | NRW a ratio | INTEG ( IF THEN ELSE (NRW a ratio < 0.01001, 0, —NRW a ratio reduction), 0.01) | - |
| 36 | NRW a ratio reduction | 0 | 1/Month |
| 37 | NRW r ratio | INTEG (IF THEN ELSE (NRW r ratio < 0.19001, 0, —NRW r ratio reduction), 0.19) | - |
| 38 | NRW r ratio reduction | 0 | 1/Month |
| 39 | NRW ratio | NRW a ratio + NRW r ratio | - |
| 40 | Population | INTEG (Population growth, 300,000) | Person |
| 41 | Population growth | Population growth rate*Population | Person/Month |
| 42 | Population growth rate | 0.001 | 1/Month |
| 43 | Real loss | NRW r ratio / (1-NRW r ratio) * "Residential & Commercial water use" | $m^3$ |
| 44 | "Residential & Commercial water use" | lpcd unit conversion * lpcd * Population | $m^3$ |
| 45 | Revenue water | (1-NRW a ratio-NRW r ratio)/(1-NRW r ratio) * "Residential & Commercial water use" | $m^3$ |
| 46 | Surface water intake | (1-Groundwater ratio) * Conveyance | $m^3$ |
| 47 | Total CF of UWS | CF of total energy + CF of water treatment chemical + CF of wastewater treatment chemical | kg$CO_2$e |
| 48 | Total Energy for UWS | Intake energy + Conveyance energy + Water treatment energy + "Transmission & Distribution energy" + End use energy + Wastewater collection energy + Wastewater treatment energy + Discharge energy + Embodied energy of water treatment chemical | kWh |

| | | + Embodied energy of wastewater treatment chemical | |
|---|---|---|---|
| 49 | Total water loss | Apparent loss + Real loss | $m^3$ |
| 50 | Total WF of UWS | Groundwater intake + Surface water intake + WF of water treatment chemical<br>+ WF of total energy + WF of wastewater treatment chemical | $m^3$ |
| 51 | "Transmission & Distribution energy" | Energy intensity td * "Transmission & Distribution" | kWh |
| 52 | "Transmission & Distribution" | Revenue water+Apparent loss+Real loss | $m^3$ |
| 53 | Wastewater chemical dosage | 20,000 | $mg/m^3$ |
| 54 | Wastewater collection | Revenue water + (Wastewater collection ratio * Apparent loss) | $m^3$ |
| 55 | Wastewater collection energy | Energy intensity wwc * Wastewater collection | kWh |
| 56 | Wastewater collection ratio | 0.9 | - |
| 57 | Wastewater ratio | 0.9 | - |
| 58 | Wastewater treatment | Wastewater ratio * Wastewater collection | $m^3$ |
| 59 | Wastewater treatment energy | Energy intensity wwt * Wastewater treatment | kWh |
| 60 | Water chemical dosage | 20,000 | $mg/m^3$ |
| 61 | Water heating energy | Energy intensity wh * Water heating ratio * "Residential & Commercial water use" | kWh |
| 62 | Water heating ratio | 0.01 | - |
| 63 | Water treatment | "Transmission & Distribution" | $m^3$ |
| 64 | Water treatment energy | Energy intensity wt * Water treatment | kWh |
| 65 | WF coef for te | 0.02 | $m^3$/kWh |
| 66 | WF coef for wtc | 1e-08 | $m^3$/mg |
| 67 | WF coef for wwtc | 1e-08 | $m^3$/mg |
| 68 | WF of total energy | WF coef for te * Total Energy for UWS | $m^3$ |
| 69 | WF of wastewater treatment chemical | WF coef for wwtc * Major chemicals for wastewater treatment | $m^3$ |
| 70 | WF of water treatment chemical | WF coef for wtc * Major chemicals for water treatment | $m^3$ |

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
