# Peer review of "Managing Apparent Loss and Real Loss from the Nexus Perspective Using System Dynamics"

_water, doi:10.3390/w14020231_

Round 1

Reviewer 1 Report

This is an interesting study. I enjoyed reading the manuscript. Nevertheless, it needs some further improvements. In general, there are still some occasional grammar errors throughout the manuscript, especially the article "the," "a," and "an" is missing in many places; please make a spellchecking in addition to these minor issues. The reviewer has listed some specific comments that might help the authors further enhance the manuscript's quality.

  1. Specific Comments

A list of acronyms is needed.

Introduction

  • The objectives should be more explicitly stated.
  • What is the novelty of this work?

Methods

  • The methodology limitation should be mentioned.
  • All variables should be explained.

Results

  • This section is well written.

Discussion

  • The discussion should summarize the manuscript's main finding(s) in the context of the broader scientific literature and address any limitations of the study or results that conflict with other published work.

Reviewer 2 Report

In this manuscript, the authors employed system dynamics to investigate water losses from the nexus perspective. The following concerns shall be addressed.

  • Nexus: In my opinion, this study underlines water-energy nexus rather than water-energy-environment nexus, as the total CO2 equivalent emissions is often seen as a classic indicator for energy sector or climate change. Yet I don’t think it is appropriate to use water-energy-climate nexus for this study either. In this regard, the authors shall add the corresponding literature survey on water-energy nexus, in particular its application to urban water system. Subsequently the authors shall elaborate which knowledge gap to be filled in this study.
  • SDG 6: The authors shall closely link the key findings from this study to SDG 6, by evincing which contributions to enabling and accelerating the progress towards achieving SDG 6, particularly for which SDG 6 target. Then in accordance with these contributions, the authors shall elaborate policy implications and its applicability to other studies.

Reviewer 3 Report

Dear authors,
thanks for your contribution firstly.
The paper provides a a novel approach for water utilities to manage water losses from the water-energy-environment Nexus perspective using system dynamics to simulate twelve scenarios with differing status of water loss and energy intensities.

The abstract is well written, it briefly summarizes the purpose of the paper and the results obtained and also the English language is fine. Overall the article is well structured.

I believe that the paragraph of the introduction is very long and that some information is redundant, furthermore I suggest you follow up on lines 57-59 with at least one bibliographic reference: "The recovery of AL is possible with little effort at a relatively low cost, and will directly improve the water utility's financial position, especially at the beginning of an NRW reduction program. " Since the text is very rich in references and on the contrary this theme (of recovery), although very important, is only touched upon, I suggest you enrich it with at least some references. In this regard, I suggest the recent publication (DOI: 10.3390/su132112318) which deals with energy recovery in water systems through the optimal positioning of micro-turbines (PAT), a topic connected to water losses, as well as to the aforementioned recovery, as it is known that through the management of the pressures in the pipeline it is possible to obtain a reduction of them.
Furthermore, even if the aim of the study is well explained it is important to highlight what the additional / new contribution to the research is.

In figures 1 and 9-12 some terms outside the boxes could be written larger in order to be more readable.

I believe that the table in the appendix has a different character from the rest of the text.

The bibliography is very large, I suggest (where possible) to prefer the recently published papers and streamline it.

In conclusion, I believe that the paper includes solid content, but the presentation need to be improved, with better structure this manuscript can have its own value and impact.

I hope that these recommendations are helpful to the authors and wish good luck for the further reviewing process.

Round 2

Reviewer 2 Report

As I mentioned in the first-round comments, this study underlines water-energy nexus rather than water-energy-environment nexus, as the total CO2 equivalent emissions is often seen as a classic indicator for energy sector or climate change. So the authors shall only use the term water-energy nexus throughout the manuscript, also the corresponding changes are expected, including the storyline, methods, results, discussion, and conclusions.

Author Response

Dear Dr. Reviewer 2,

At the moment that we are preparing for the second review report, uploading a new manuscript was not available. Thus, the review report and the edited manuscript were combined and uploaded.

Sincerely,

Dr. Eunher Shin

Reviewer 3 Report

Dear Authors,

the suggested changes have been made. Now, in my opinion, the paper is ready for publication.

Author Response

Dear Dr. Reviewer,

We appreciate the time and effort you have dedicated to providing insightful feedback on ways to strengthen our paper.

Thank you for your consideration.

Sincerely,

Dr. Eunher Shin
Research and Development Division, UNESCO International Centre for Water Security and Sustainable Management
Daejeon 34045, Republic of Korea
Tel: +82-42-349-0007
E-mail: [email protected]

Round 3

Reviewer 2 Report

I am very happy that the authors addressed my concerns. No further comments from my side.